# Molecular-Charge-Contact-Based Ion-Sensitive Field-Effect Transistor Sensor in Microfluidic System for Protein Sensing

**DOI:** 10.3390/s19153393

**Published:** 2019-08-02

**Authors:** Haoyue Yang, Toshiya Sakata

**Affiliations:** Department of Materials Engineering, School of Engineering, The University of Tokyo, 7-3-1 Hongo, Bunkyo-ku, Tokyo 113-8656, Japan

**Keywords:** ion-sensitive field-effect transistor, molecular charge contact, streptavidin–biotin interaction, pH, microfluidics

## Abstract

In this paper, we demonstrate the possibility of direct protein sensing beyond the Debye length limit using a molecular-charge-contact (MCC)-based ion-sensitive field-effect transistor (ISFET) sensor combined with a microfluidic device. Different from the MCC method previously reported, biotin-coated magnetic beads are set on the gate insulator of an ISFET using a button magnet before the injection of target molecules such as streptavidin. Then, the streptavidin—a biotin interaction, used as a model of antigen—antibody reaction is expected at the magnetic beads/gate insulator nanogap interface, changing the pH at the solution/dielectric interface owing to the weak acidity of streptavidin. In addition, the effect of the pH or ionic strength of the measurement solutions on the electrical signals of the MCC-based ISFET sensor is investigated. Furthermore, bound/free (B/F) molecule separation with a microfluidic device is very important to obtain an actual electrical signal based on the streptavidin–biotin interaction. Platforms based on the MCC method are suitable for exploiting the advantages of ISFETs as pH sensors, that is, direct monitoring systems for antigen–antibody reactions in the field of in vitro diagnostics.

## 1. Introduction

A solution-gate ion-sensitive field-effect transistor (ISFET) was proposed for detecting ions in biological environments [1]. In this device, electrolyte solutions were assumed to induce the interfacial potential between the solution and the gate insulator instead of a metal gate in a metal-oxide-semiconductor (MOS) transistor, although it was necessary to use a reference electrode in the solution. A gate insulator is often composed of oxide or nitride membranes such as Ta_2_O_5_, Al_2_O_3_, Si_3_N_4_, and SiO_2_; therefore, hydroxyl groups at the oxide or nitride surface in a solution undergo the equilibrium reaction with hydrogen ions through protonation (–OH + H^+^ ⇄ –OH_2_^+^) and deprotonation (–OH ⇄ –O^−^ + H^+^). Thus, the change in the surface charge is detected from the change in pH based on the principle of the field effect [2,3,4]. Moreover, the concept of direct immunosensing using an ISFET sensor was proposed for the label-free monitoring of antigen–antibody reactions that overcomes the shortcomings of a classical immunoassay [5]. In the case of an immuno-ISFET sensor, antibodies are mostly immobilized on the gate insulator to selectively detect antigens on the basis of intrinsic molecular charges, but counter ions shield these charges from the solution/dielectric interface because antibodies are often large. That is, electrolyte solutions include equal numbers of positive and negative ions, which results in a neutral solution. On the other hand, a deviation of the positive or negative charge density is found in the vicinity of the gate insulator depending on the density of the surface charges, which form a diffusion layer on the gate insulator, the thickness of which is defined as the Debye length [6]. The Debye length in a physiological solution has been calculated to be less than 1 nm. This means that a large antibody (> 1 nm) will react with an antigen located at a distance exceeding the Debye length, which cannot be detected by an immuno-ISFET sensor [7,8,9]. The detection limit, that is, the Debye length limit, of immuno-ISFET sensors has affected the development of the entire range of DNA-based ISFET sensors for single-nucleotide polymorphism (SNP) genotyping and DNA sequencing based on intrinsic molecular charges [10,11,12,13,14,15,16,17,18].

The molecular charge contact (MCC) method is reasonable for biomacromolecular recognition, including not only DNA extension reactions [19] but also antibody—antigen reactions with an ISFET-based biosensor. Receptor biomolecules such as DNA probes and antibodies are immobilized on magnetic beads, to which target molecules in a sample solution are selectively bound, as shown in Figure 1a. Then, the magnetic beads are made to approach the gate insulator of the ISFET in a measurement solution using a magnet at the bottom of the device. In this case, a change in the molecular charges on the magnetic beads is induced on the gate insulator of the ISFET and then detected as the difference between the signals before and after the biomolecular recognition events on the magnetic beads, corresponding to the shift in the electrical signals based on the target biomolecules—owing to a bound/free (B/F) molecule separation. In general, biomolecular recognition events proceed on the gate insulator of an ISFET. However, molecular charges away from the gate insulator are shielded by counter ions, depending on the ionic strength (the Debye length limit). This is why large probe biomolecules hardly induce any change in the density of molecular charges upon their interaction with target biomolecules such as antigen—antibody reactions in biological samples, which react with such probes over a distance exceeding the Debye length. Therefore, the MCC method is reliable for use with the methodology of ISFET-based biosensors, because antigens, which are bound to antibodies on the magnetic beads, can be detected by for instance making them directly approach the gate insulator. However, background noise generated by the injection of magnetic beads, which corresponds to the effect of the direct attachment of magnetic beads at the gate insulator, distorts the actual signal used for biomolecular recognition. This phenomenon occurs twice, before and after the reaction of probe molecules with the target biomolecules (Figure 1b and Appendix A). Therefore, the magnetic beads with probe biomolecules are set on the gate insulator with a magnet in advance (Figure 1c). After that, target biomolecules are added to the interface between the magnetic beads with probe biomolecules and the gate insulator, keeping the beads attached to the magnet. As a result, the change in the density of molecular charges, based on the probe—target interaction at the interface, is detected as the electrical signal of the ISFET (Figure 1c). That is, the effect of the injected beads on the electrical signal is negligible. This improved MCC methodology is based on the detection principle of a cultured cell—gate ISFET sensor, which enables the detection of the change in pH at the cell/gate insulator nanogap interface caused by cellular respiration [20,21].

In this study, we have improved the MCC method for ISFET biosensors. In particular, a microfluidic system is employed to measure a biological reaction in real time. Additionally, the streptavidin—biotin interaction is utilized as a model of antigen—antibody reactions. Streptavidin comprises four subunits and has a molecular weight of about 60,000. Streptavidin—biotin complexes are useful in a wide range of biotechnological applications such as tagging and molecule delivery.

## 2. Materials and Methods

### 2.1. Reagents

The chemicals used were obtained from the following sources: PureCube Biotin MagBeads XL from Cube Biotech, Monheim am Rhein, Germany; streptavidin from FUJIFILM Wako Pure Chemical Corporation, Osaka, Japan; 1X phosphate-buffered saline (10010-PBS, pH 7.4, and pNa 0.8) from Thermo Fisher Scientific, Waltham, MA, USA; and standard buffer solutions [pHs 4.01 (50 mM C_6_H_4_(COOK)(COOH)) and 10.01 (25 mM NaHCO_3_ and 25 mM Na_2_CO_3_)] from Wako.

### 2.2. Pretreatment of Magnetic Beads

In accordance with the gate size of the ISFET used in this study, magnetic beads with 90 μm diameter were employed in this study. As the magnetic beads were delivered in a low-salt buffer with 20% ethanol, thorough washing was necessary before electrical measurements. Magnetic beads (5 μL) in a stock solution were transferred into a 1.5 mL microcentrifuge tube with 50 μL deionized water and then magnetically captured on the sidewall using a magnetic stand (Thermo Fisher). The supernatant was aspirated, and the magnetic beads were washed three times with 100 μL measurement solutions (1X PBS or other measurement solutions). After washing, the magnetic beads were resuspended in a 50 μL measurement solution. Each 10 μL pretreated solution included 1 μL magnetic beads, that is, the number of magnetic beads was about 100.

### 2.3. Detection of Streptavidin by MCC Method with Microfluidic System

A microfluidic system was designed for this measurement. As shown in Figure 2, four holes were made in an acrylic plate for the inlet (first hole) and outlet (fourth hole) to allow the flow of a measurement solution, for the injection of reagents (second hole), and for the insertion of a reference electrode (third hole). An ISFET sensor was set exactly under the second hole on an acrylonitrile butadiene styrene (ABS) plate, and a button magnet was placed under the sensor area to fix magnetic beads on the gate insulator. The ISFET sensors (ISFETCOM Co., Ltd., Saitama, Japan) used in this study were composed of a silicon-based n-channel depletion-mode FET with a Ta_2_O_5_/SiO_2_ (100 nm/50 nm) layer as a gate insulator with a width (*W*) and length (*L*) of 340 and 10 μm, respectively. The Ta_2_O_5_ thin film was used as a passivation layer to prevent the leakage of current into the buffer solution. The gate voltage (*V*_G_)—drain current (*I*_D_) electrical characteristics were measured using a semiconductor parameter analyzer (B1500A, Agilent Technologies, Santa Clara, CA, USA). The change in *V*_G_ in the *V*_G_–*I*_D_ electrical characteristics was estimated as the threshold voltage (*V*_T_) shift, which was evaluated at a constant *I*_D_ of 1 mA and a constant drain voltage (*V*_D_) of 2.5 V. Moreover, the interfacial potential at the solution/dielectric interface (*V*_out_) was monitored in real time using a circuit with which the change in the solution/dielectric interfacial potential (Δ*V*_out_) can be read out directly at a constant *I*_D_ (Appendix A). In this study, *V*_D_ and *I*_D_ were set to be 2.5 V and 1 mA, respectively. An Ag/AgCl reference electrode was directly immersed in the measurement solution through the third hole on the fluidic structure. As the measurement solution, which was allowed to flow from the inlet to the outlet, 1X PBS was also diluted 10-fold (0.1X) and 100-fold (0.01X) to examine the effect of the ionic strength on the electrical signal of the ISFET sensor. In addition, the standard buffer solutions (pHs 4.01 and 10.01) were utilized to examine the effect of the pH on the electrical signal of the ISFET sensor.

To cover the gate insulator of the ISFET with magnetic beads, 10 μL of the pretreated magnetic bead solution was gently added dropwise through the second hole, ensuring full coverage on the gate insulator. Then, a buffer solution was allowed to flow from the inlet at a rate of 74 μL/min using a pumping system (GILSON) until Δ*V*_out_ became stable (stage 1). After stopping the flow, 20 μL of streptavidin solution was injected into the measurement system through the second hole. The concentration of streptavidin in the 0.1X PBS solution was varied from 0.18 mM to 1.8 mM. To ensure streptavidin—biotin binding at the magnetic beads and the gate insulator interface, data were collected for about 60 min (stage 2). Finally, the flow system was operated to wash out unreacted streptavidin until Δ*V*_out_ became stable (stage 3). That is, such a biomolecular recognition is believed to have been realized by B/F molecule separation. The difference between $\Delta V_{\mathrm{out}}^{\mathrm{stage}2}$ at stage 2 and $\Delta V_{\mathrm{out}}^{\mathrm{stage}1}$ at stage 1 was estimated and denoted as Δ*V*_1_ (= $\Delta V_{\mathrm{out}}^{\mathrm{stage}2}-\Delta V_{\mathrm{out}}^{\mathrm{stage}1}$), and the difference between $\Delta V_{\mathrm{out}}^{\mathrm{stage}3}$ at stage 3 and $\Delta V_{\mathrm{out}}^{\mathrm{stage}1}$ at stage 1 was estimated and denoted as Δ*V*_2_ (= $\Delta V_{\mathrm{out}}^{\mathrm{stage}3}-\Delta V_{\mathrm{out}}^{\mathrm{stage}1}$).

## 3. Results and Discussion

### 3.1. Potentiometric Detection of Streptavidin Using MCC Method in Microfluidic System

Figure 3 shows the real-time measurement of streptavidin using the MCC method in the microfluidic system. The 0.1X PBS solution was used for the measurements, in which the concentration of streptavidin was 180 μM. Biotin-coated magnetic beads were fixed on the gate insulator with the magnet. The number of magnetic beads on the gate insulator was about seven. As shown in Figure 3, Δ*V*_out_ for the ISFETs both with and without magnetic beads increased upon the injection of the streptavidin solution and subsequently saturated. Then, the potential for both ISFETs decreased after washing the magnetic beads. Considering the change in the electrical signal of the ISFET without the magnetic beads, for every measurement Δ*V*_1_ included the pH response to the injection of the streptavidin solution. In fact, the 0.1X PBS solution that included streptavidin (180 μM) showed a pH of 5.1, which was a decrease from pH 6.6 for the original 0.1X PBS solution. However, Δ*V*_2_ was almost zero for the ISFETs without magnetic beads (control measurements), which is to say that the pH responses were canceled out by the washing process, whereas Δ*V*_2_ for the ISFET with the biotin-coated magnetic beads shifted in the positive direction by about 5 mV (sample measurement). This was due to the streptavidin–biotin interaction at the magnetic beads/gate interface. In other words, Δ*V*_1_ was an unclear response owing to the effect of dissolved streptavidin on the change in pH in the sample solution, whereas Δ*V*_2_ was a reliable response for the streptavidin—biotin interaction because of the B/F molecule separation and the electrical measurement in the buffer solution used as the measurement solution before and after the reaction.

Moreover, Δ*V*_2_ for the sample measurement increased with the decreasing ionic strength of the PBS solution from 1X to 0.01X, close to a neutral pH (pH 6.0 to 7.4) (Figure 4a), although Δ*V*_2_ obtained by the control measurement hardly changed except for the measurement in the 0.01X PBS solution (Appendix A). In fact, the Debye length at a solution/gate interface based on the electrical double layer (EDL) structure strongly depends on the ionic strength of the electrolyte solution, as expressed by:λ = (*ε*_0_*ε*_r_*k*_B_*T*/2*N*_A_*e*^2^*I*)^1/2^(1)
where *I* is the ionic strength of the electrolyte, *ε*_0_ is the permittivity of free space, *ε*_r_ is the dielectric constant, *k*_B_ is the Boltzmann constant, *T* is the absolute temperature, *N*_A_ is the Avogadro number, and *e* is the elementary charge [22]. According to Equation (1), the Debye length increases with a decreasing ionic strength. Thus, the increase in Δ*V*_2_ in the sample measurement appears to have been caused by the decrease in the ionic strength of the measurement solution, which resulted in the increase in the Debye length at the solution/dielectric interface.

Moreover, Δ*V*_2_ in the sample measurement was examined while varying the pH in the measurement solution. Figure 4b shows Δ*V*_2_ obtained using the biotin-coated magnetic beads in the measurement solutions with pHs 4.01 and 10.01. Here, measurement solutions with almost the same ionic strength but different pHs should be used. Δ*V*_2_ was about 20 mV when measured at pH 10.01, which was larger than that measured at pH 4.01. The isoelectric point (pI) of streptavidin is known to be ~5.6 [23,24]. In other words, the total net charge of streptavidin is assumed to have been negative at pH 10.01. However, considering the source follower circuit used to output the interfacial potential at the solution/dielectric interface (Appendix A), Δ*V*_2_ would have shifted in the negative direction if the detected electrical signals were based on the negative charges of biomolecules. In other words, the positive shift of Δ*V*_2_ obtained in this study may have depended on a different mechanism to the Debye length limit, which is discussed in Section 3.3.

### 3.2. Limit of Detection

As mentioned above, Δ*V*_2_ was used to estimate the concentration of detected streptavidin using the MCC method. As shown in Figure 5, Δ*V*_2_ in the sample measurement was evaluated from single logarithmic plots of the streptavidin concentrations from 0.18 μM to 1.8 mM in the 0.1X PBS solution. The concentration range within which streptavidin can be detected was assumed to be roughly from 1.8 μM to 180 μM, where the detection sensitivity was about 5 mV/decade. Using the Kaiser limit [25], the limit of detection (LOD) for streptavidin using the MCC method was calculated to be about 2.3 μM (Appendix A). The calculated LOD is equal or inferior to the LODs obtained in previous studies, as shown in Appendix A [26,27,28,29,30]. However, the detection of biomacromolecules interacting with receptor molecules tethered at the gate insulator of an ISFET is not possible owing to the Debye length limit [7,8,9]. That is, the signals detected with the MCC method provide the possibility of directly detecting biomacromolecules beyond the Debye length limit using an ISFET sensor, although the LOD should be improved in the future.

### 3.3. Detection Mechanism

In this study, the biotin-coated magnetic beads were set up on the gate insulator using a button magnet before the injection of the streptavidin solution as a sample. As hydrogen ions or histamine molecules released from cells were concentrated in the cell/gate insulator nanogap interface, where the change in pH was induced [20,21,31], the injected streptavidin molecules may have flowed into the magnetic beads/gate insulator nanogap interface and then contributed to the change in pH there. Streptavidin molecules are weakly acidic; thus, the interfacial pH in the magnetic beads/gate insulator nanogap interface is expected to decrease after the streptavidin–biotin interaction. This can also be explained from the signal direction. Δ*V*_2_, shown in Figure 3, shifted in the positive direction, which means that the density of positive charges increased or the density of negative charges decreased at the solution/dielectric interface. Eventually, the weakly acidic streptavidin molecules would have brought hydrogen ions with positive charges into the magnetic beads/gate insulator nanogap interface. As a result, the ISFET sensor would have detected the change in pH on the basis of the equilibrium reaction between hydrogen ions and hydroxyl groups at the oxide gate insulator. Basically, the ISFET sensors used in this study showed a sensitivity of about 55 mV/pH [32], as shown in Appendix A. In accordance with the electrical properties of the ISFET sensors, the streptavidin–biotin interaction on the magnetic beads was assumed to be detected as the change in interfacial pH at the magnetic beads/gate insulator nanogap interface by the MCC method. This means that a previously reported biologically coupled gate FET (bio-FET) sensor [7,8,9,10,11,12,13,14,15,16,17,18], whose gate insulator was chemically modified with probe molecules, did not enable such a streptavidin detection. That is, the MCC method is suitable for exploiting the advantages of ISFETs as pH sensors. In addition, more hydrogen ions would have been dissociated from streptavidin molecules at the magnetic beads/gate insulator nanogap interface in the measurement solution with pH 10.01 (alkaline solution), resulting in the larger positive shift in the solution/dielectric interfacial potential of the ISFET (Figure 4b). Furthermore, the lower the ionic strength of the measurement solution, the larger the value of Δ*V*_2_ for the ISFET sensor, as shown in Figure 4a. This is considered to be due to the observation that the effect of counter ions such as Na^+^ on hydroxy groups at the oxide gate insulator was smaller in a measurement solution with a lower ionic strength. As a result, more hydrogen ion—hydroxy group interactions should have been induced at the oxide gate insulator. Δ*V*_2_ was estimated in the same measurement solution before and after biomolecular recognition events. Therefore, Δ*V*_2_ can be amplified by controlling the pH and ionic strength of the measurement solution, although the electrical drift (stability) of the ISFET sensor and the denaturation of target proteins should be considered for a variety of measurement solutions. In particular, it is necessary to exclude the effects of impurities and to control the measurement conditions for the microfluidic system.

## 4. Conclusions

A problem of bio-FET sensors has been that biomacromolecules such as proteins cannot be detected using probe-tethered gate ISFET sensors because their charges are easily shielded by counter ions in sample solutions. This is well known as the Debye length limit. Indeed, the Debye screening effect on the biotin—streptavidin interaction at the sensor substrate was investigated previously [24], showing that the electrical signal of nanowire-FET devices was suppressed by the Debye screening in a 1X PBS solution, compared with that in the diluted buffer 0.01X PBS solution. Therefore, the 1 mM sodium bicarbonate buffer solution (pH 8.4), whose ionic strength was near that of the 0.01X PBS solution, was utilized for streptavidin–biotin sensing to avoid the Debye screening effect reported in a previous paper [23]. However, the drift of electrical signals in the 0.01X PBS solution may not have been negligible owing to the deteriorating buffer effect. That is, the 0.1X PBS solution was employed for the real-time measurement (Figure 3) and LOD analysis (Figure 5) in this study. Moreover, to avoid the Debye screening effect, the MCC method can be used for the detection of biomacromolecular recognition events such as antigen—antibody reactions using an ISFET sensor, because targeted biomolecules at magnetic beads are in direct contact with the gate insulator. In fact, the effect of counter ions on the electrical signals of an MCC-based ISFET sensor may not be negligible, but the change in pH at the magnetic beads/gate insulator nanogap interface was a key point in detecting streptavidin–biotin interactions in this study. This resulted from the setting of the magnetic beads on the gate insulator using a magnet placed below the substrate before the injection of streptavidin as a sample. In other words, the chemical rationale for the electrical signals obtained in this study appears to be similar to that based on the change in pH at the cell/gate insulator nanogap interface in previous studies. Although the LOD obtained in this study was inferior to the LODs obtained in previous studies, it can be improved by regulating the density of immobilized probe molecules such as biotin on magnetic beads, the size of the magnetic beads and the gate insulator, and so forth. Considering the above, we have clarified the possibility of the direct detection of the streptavidin–biotin interaction as a model of antigen—antibody reactions using an MCC-based ISFET sensor. Platforms based on the MCC method are suitable for exploiting the advantages of ISFETs as pH sensors and are expected to be applied to direct monitoring systems for antigen—antibody reactions in the field of in vitro diagnostics.

## Figures and Tables

**Figure 1 sensors-19-03393-f001:**
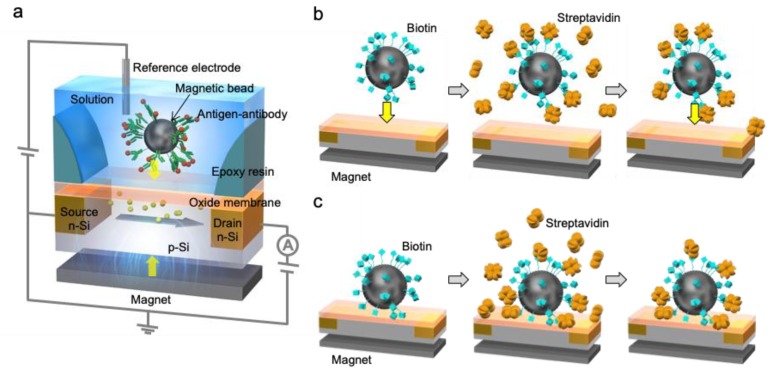
Schematic illustration of the molecular charge contact (MCC)-based ISFET sensor. (**a**) Basic structure of electrical measurement using the MCC-based ISFET sensor. Biomolecular recognition events such as an antigen—antibody reaction occur on magnetic beads but not at the gate insulator of the ISFET. Magnetic beads with biomolecules are made to approach the gate insulator using a button magnet from below the ISFET substrate; (**b**) the previously reported MCC method; and (**c**) the improved MCC method in this study.

**Figure 2 sensors-19-03393-f002:**
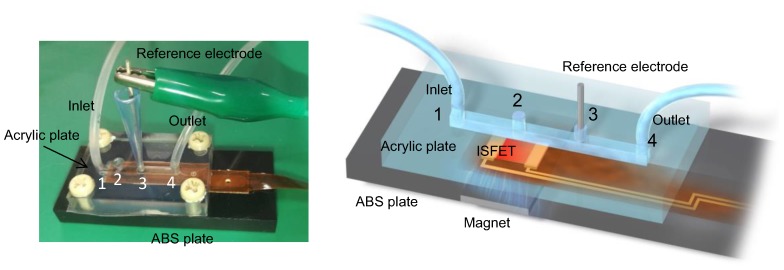
The microfluidic system including the MCC-based ISFET sensor. Four holes were made in an acrylic plate for the inlet (first hole) and outlet (fourth hole) to allow the flow of a measurement solution, for the injection of reagents (second hole), and for the insertion of a reference electrode (third hole). An ISFET sensor was set exactly under the second hole on an acrylonitrile butadiene styrene (ABS) plate, and a button magnet was placed under the sensor area to fix magnetic beads on the gate insulator.

**Figure 3 sensors-19-03393-f003:**
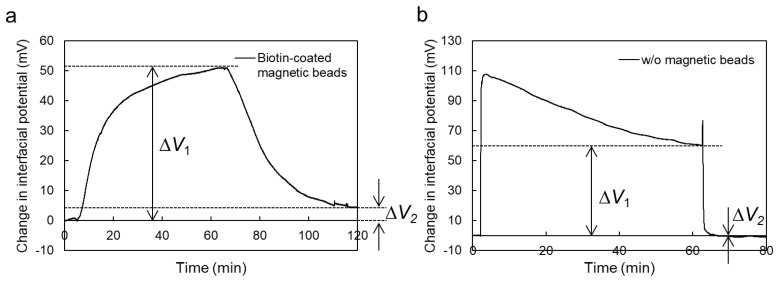
Real-time measurement of change in interfacial potential at solution/dielectric interface of MCC-based ISFET sensor. Biotin-coated magnetic beads were utilized for the detection of streptavidin using the MCC-based ISFET sensor [(**a**) sample measurement], whereas an ISFET without magnetic beads was also prepared to estimate the effect of background noises such as the change in pH on the electrical signals of the MCC-based ISFET sensor [(**b**) control measurement]. The volume and concentration of the injected streptavidin solution were 20 μL and 180 μM, respectively.

**Figure 4 sensors-19-03393-f004:**
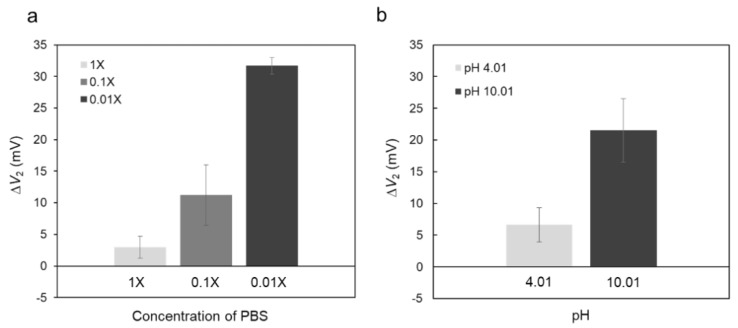
The effect of the electrolyte solution used for the electrical measurement on the electrical signal of the MCC-based ISFET sensor. (**a**) Δ*V*_2_ at different ionic strengths. 0.01X, 0.1X, and 1X PBS solutions were used as the measurement solutions. Each signal was averaged over three measurements (n = 3) with the standard deviation shown as error bars; (**b**) Δ*V*_2_ for different pHs. pHs 4.01 and 10.01 buffer solutions were used for the measurement solutions. Each signal was averaged over three measurements (n = 3) with the standard deviation shown as error bars.

**Figure 5 sensors-19-03393-f005:**
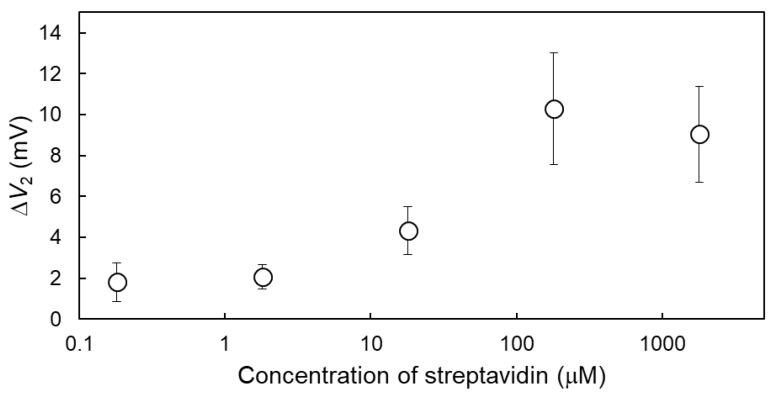
Δ*V*_2_ at various streptavidin concentrations. The concentration of streptavidin was varied from 0.18 μM to 1.8 mM in the 0.1X PBS solution. Each signal was averaged over three measurements (n = 3) with the standard deviation shown as error bars.

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
