# Peer review of "Molecular-Charge-Contact-Based Ion-Sensitive Field-Effect Transistor Sensor in Microfluidic System for Protein Sensing"

_sensors, 2019, doi:10.3390/s19153393_

Round 1

Reviewer 1 Report

@page { margin: 0.79in } p { margin-bottom: 0.1in; line-height: 120% }

The manuscript entitled “Molecular-Charge-Contact-Based Ion-Sensitive Field-Effect Transistor Sensor in Microfluidic System for Protein Sensing” deals with magnetic beads, functionalized to enable selective biosensing, suspended in an electrolyte solution. Different from the Molecular-Charge-Contact methods previously reported, biotin-coated magnetic beads are set on the dielectric surface of an ISFET using a button magnet before the injection of target molecules such as streptavidin. Then, the streptavidin–biotin interaction as a model of antigen–antibody reactions is expected at the magnetic beads/gate nanogap interface, which changes the pH at the gate owing to the weak acidity of streptavidin. With this strategy, direct protein sensing beyond the Debye

length limit using a molecular charge contact (MCC)-based ion-sensitive field-effect transistor

(ISFET) is proposed.

I would like to congratulate the authors for the refreshing work and careful analysis of the presented data. The manuscript is well written and definitively suitable for publication in Sensors.

There are, however, a few small minor comments I would like to suggest to make the manuscript more easy to understand for an even broader readership.

1) Page1), line 28:

In this device, electrolyte solutions were assumed to induce the interfacial potential

between the solution and the gate insulator instead of a metal gate in a metal-oxide-semiconductor

(MOS) transistor, although it was necessary to use a reference electrode in the solution.

I agree that “electrolyte” gated transistors can be approximated to some instance with conventional MOS transistors. I think, though, that the “reference” electrode as used in the manuscript is misleading. A reference electrode is typically used to “measure” a voltage, while in this case actually its connected in series with the gate dielectric through the electrolyte solution and acts essentially as gate-electrode. Choosing the Ag/AgCl electrode is a very smart move, since it reduces the potential drop at the gate/electrolyte interface and shifts the potential drop towards the electrolyte/dielectric/semiconductor interface. I think it is very important to distinguish between the “gate” which typically is a conductor or even metal in ISFETs from the dielectric interface, which is an insulator.

Therefore I suggest to change “gate surface”, attachment to the “gate” and similar expressions throughout the manuscript to “solution/dielectric interface”, “active sensing interface” or something similar.

2) Supporting Information, page 3, Figure 2:

The change of Vout or DeltaV2 at different concentrations of PBS cached my eye and is very interesting. The ISFET sensor platform shown with SiO2/Ta2O5 as active sensing interface is highly sensitive to pH and very close to Nernst limit with a sensitivity of 55mV/pH. When diluting PBS1x down to PBS0.01x a clear negative shift of Vth of about 20 mV was reported. This would translate to a pH change of about: 0.36 pH (20mV / 55 mV/pH). Since I guess that dilution was done with “pure” water under ambient conditions, dissolution CO2 probably makes the PBS0.01x more acidic compared to PBS 1x and PBS0.1x. Due to the high pH sensitivity of the dielectric surface this could be resolved and measured.

I think its good to include or discuss this topic further and would add more strength to the conclusion of the paper.

3) Minor aspects, check expressions, sometimes the voltage shift is shown as V, sometimes as DeltaV, unify.

4) For completeness, maybe its good to add also the buffer composition of the solutions in Materials and Methods.

With that I would like to congratulate the authors again for this nice work!

Author Response

Dear Editor:

Thank you very much for your kind reviews. We have revised our manuscript in accordance with the reviewers’ comments as follows. We would be grateful if the manuscript could be re-valuated for publication in Sensors.

[Reviewer: 1]

Comments and Suggestions for Authors:

The manuscript entitled “Molecular-Charge-Contact-Based Ion-Sensitive Field-Effect Transistor Sensor in Microfluidic System for Protein Sensing” deals with magnetic beads, functionalized to enable selective biosensing, suspended in an electrolyte solution. Different from the Molecular-Charge-Contact methods previously reported, biotin-coated magnetic beads are set on the dielectric surface of an ISFET using a button magnet before the injection of target molecules such as streptavidin. Then, the streptavidin–biotin interaction as a model of antigen–antibody reactions is expected at the magnetic beads/gate nanogap interface, which changes the pH at the gate owing to the weak acidity of streptavidin. With this strategy, direct protein sensing beyond the Debye length limit using a molecular charge contact (MCC)-based ion-sensitive field-effect transistor (ISFET) is proposed.

I would like to congratulate the authors for the refreshing work and careful analysis of the presented data. The manuscript is well written and definitively suitable for publication in Sensors.

There are, however, a few small minor comments I would like to suggest to make the manuscript more easy to understand for an even broader readership.

1) Page1), line 28:

In this device, electrolyte solutions were assumed to induce the interfacial potential between the solution and the gate insulator instead of a metal gate in a metal-oxide-semiconductor (MOS) transistor, although it was necessary to use a reference electrode in the solution.

I agree that “electrolyte” gated transistors can be approximated to some instance with conventional MOS transistors. I think, though, that the “reference” electrode as used in the manuscript is misleading. A reference electrode is typically used to “measure” a voltage, while in this case actually its connected in series with the gate dielectric through the electrolyte solution and acts essentially as gate-electrode. Choosing the Ag/AgCl electrode is a very smart move, since it reduces the potential drop at the gate/electrolyte interface and shifts the potential drop towards the electrolyte/dielectric/semiconductor interface. I think it is very important to distinguish between the “gate” which typically is a conductor or even metal in ISFETs from the dielectric interface, which is an insulator.

Therefore I suggest to change “gate surface”, attachment to the “gate” and similar expressions throughout the manuscript to “solution/dielectric interface”, “active sensing interface” or something similar.

(Reply)

Thank you very much for the kind review. In accordance with the reviewer’s comment, we have carefully changed the expressions “gate surface” and “gate” to “solution/dielectric interface” and “gate insulator”, respectively, in the revised manuscript. 

2) Supporting Information, page 3, Figure 2:

The change of Vout or DeltaV2 at different concentrations of PBS cached my eye and is very interesting. The ISFET sensor platform shown with SiO2/Ta2O5 as active sensing interface is highly sensitive to pH and very close to Nernst limit with a sensitivity of 55mV/pH. When diluting PBS1x down to PBS0.01x a clear negative shift of Vth of about 20 mV was reported. This would translate to a pH change of about: 0.36 pH (20mV / 55 mV/pH). Since I guess that dilution was done with “pure” water under ambient conditions, dissolution CO2 probably makes the PBS0.01x more acidic compared to PBS 1x and PBS0.1x. Due to the high pH sensitivity of the dielectric surface this could be resolved and measured.

I think its good to include or discuss this topic further and would add more strength to the conclusion of the paper.

(Reply)

In accordance with the reviewer’s suggestion, we have added the following sentence in the revised Supplementary Materials.

“On the other hand, the diluted buffer solutions, such as 0.01X PBS, might have been affected by the dissolution of carbon dioxide in them under ambient conditions.” (Supplementary Materials)

3) Minor aspects, check expressions, sometimes the voltage shift is shown as V, sometimes as DeltaV, unify.

(Reply)

In accordance with the reviewer’s comment, we have unified the description of the voltage shift as DV.

4) For completeness, maybe its good to add also the buffer composition of the solutions in Materials and Methods.

(Reply)

We have already described some information about 1X PBS such as its model number, pH, and pNa in Materials and Methods (2.1. Reagents), which are considered to roughly indicate ionic strengths in the diluted buffer solutions. Moreover, in accordance with the reviewer’s suggestion, we have added the buffer compositions as follows.

“pHs 4.01 (50 mM C6H4(COOK)(COOH)) and 10.01 (25 mM NaHCO3and 25 mM Na2CO3)” (page 3, line 95)

With that I would like to congratulate the authors again for this nice work!

- Open Review

( ) I would not like to sign my review report 

(x) I would like to sign my review report 

- English language and style

( ) Extensive editing of English language and style required 

(x) Moderate English changes required 

( ) English language and style are fine/minor spell check required 

( ) I don't feel qualified to judge about the English language and style 

- Does the introduction provide sufficient background and include all relevant references?

Yes

- Is the research design appropriate?

Yes

- Are the methods adequately described?

Can be improved

Are the results clearly presented?

Can be improved

- Are the conclusions supported by the results?

Yes

Reviewer 2 Report

The authors show that the Debye shielding limit can be surpassed by monitoring the charge rearrangements and pH level accompanying attachments of target molecules to the fixed sensor molecules on the gate of ISFET. 

Author Response

Dear Editor:

Thank you very much for your kind reviews. We have revised our manuscript in accordance with the reviewers’ comments as follows. We would be grateful if the manuscript could be re-valuated for publication in Sensors.

[Reviewer: 2]

Comments and Suggestions for Authors:

The authors show that the Debye shielding limit can be surpassed by monitoring the charge rearrangements and pH level accompanying attachments of target molecules to the fixed sensor molecules on the gate of ISFET.

(Reply)

Thank you very much for the kind review. 

 - Open Review

(x) I would not like to sign my review report 

( ) I would like to sign my review report 

- English language and style

( ) Extensive editing of English language and style required 

( ) Moderate English changes required 

(x) English language and style are fine/minor spell check required 

( ) I don't feel qualified to judge about the English language and style 

- Does the introduction provide sufficient background and include all relevant references?

Yes 

- Is the research design appropriate?

Yes

- Are the methods adequately described?

Yes

- Are the results clearly presented?

Yes

- Are the conclusions supported by the results?

Yes

Reviewer 3 Report

Dear Editor,

I carefully revised the paper “Molecular-Charge-Contact-Based Ion-Sensitive Field-Effect Transistor Sensor in Microfluidic System for Protein Sensing” by Yang and Sakata. The paper reports about the employment of MCC for the development of ISFET-based immunosensors. Biotin-streptavidin assay has been used for demonstrating the device functionality. In particular, authors claim that superior device performances are obtained by performing the approach of magnetic beads before biotin-streptavidin interaction, and that the setup is capable to operate beyond the Debye length.

I don’t find the paper suitable for publication in Sensors. I’m basing my opinion on the following aspects:

the quality of the presentation is low. English level is not sufficient, several periods are definitely confused and misleading, in particular in the introduction. There are several conceptual errors in the description of ISFETs (biochemical receptors are not anchored on the gate, but on the gate insulator! The sensing surface is those of the gate insulator, and not that of the gate).

Control experiments are not properly described in the main paper, only a few informations are provided in the Supplementary Information. It is not clear how an aspecific absorption can represent a control measurement in this setup. A direct immobilization of biotin-streptavidin would be more significant, as authors can demonstrate the Debye screening would happen in that case.

Authors state that their modification of the MCC approach, i.e. the magnetic anchoring of beads before biotin-streptavidin interaction, is more effective than two-stage magnetic bead approach. Why don’t reporting a demonstration of this?

Authors would pay attention to the fact that their LOD (limit of detection, and NOT limitation on detection) is HIGHER than those reported in liturature, and not lower. They are considering this definition in an opposite way than the one normally accepted in literature. Moreover, 4 references are sufficient in the very wide range of publication about immunoFETs, and also the “Kaiser method” reported in the paper would be described, at least in SI as the related reference is in german!

The explanation of Figure 4b is not clear. If streptavidin has a pI of 5.6, it should protonate at pHpI. Therefore, if pH is swept from acid to basic values, the positive charge would increase, and so the n-type ISFET would be switched off. Also considering an indirect detection through pH changes in the bead/gate insulator nanogap, this would be monotonic. So, it is not clear why DV2 is first decreasing (from 4.01 to 7.41) and then increasing (from 7.41 to 10.01). The explanation provided in Section 3.3 is confused and, according to the Reviewer, is more an hyphotesis (badly discussed) than a proof.

Too much emphasis is put in the fact that the ISFET has been integrated with a microfluidic. It is not the first time, and the proposed microfluidic is something very standard (I can say also very rudimentary).

According to these consideration, I suggest rejection.

Author Response

Dear Editor:

Thank you very much for your kind reviews. We have revised our manuscript in accordance with the reviewers’ comments as follows. We would be grateful if the manuscript could be re-valuated for publication in Sensors.

[Reviewer: 3]

Comments and Suggestions for Authors:

I carefully revised the paper “Molecular-Charge-Contact-Based Ion-Sensitive Field-Effect Transistor Sensor in Microfluidic System for Protein Sensing” by Yang and Sakata. The paper reports about the employment of MCC for the development of ISFET-based immunosensors. Biotin-streptavidin assay has been used for demonstrating the device functionality. In particular, authors claim that superior device performances are obtained by performing the approach of magnetic beads before biotin-streptavidin interaction, and that the setup is capable to operate beyond the Debye length.

I don’t find the paper suitable for publication in Sensors. I’m basing my opinion on the following aspects: the quality of the presentation is low. English level is not sufficient, several periods are definitely confused and misleading, in particular in the introduction. There are several conceptual errors in the description of ISFETs (biochemical receptors are not anchored on the gate, but on the gate insulator! The sensing surface is those of the gate insulator, and not that of the gate).

(Reply)

Thank you very much for the careful review. In accordance with the reviewer’s comment, we have corrected the conceptual errors in the description of ISFETs, that is, we have changed the expression “the gate” to “the gate insulator” in the revised manuscript. Also, we have revised the presentation of the entire manuscript to improve its quality, taking the comments of all the reviewers into consideration. 

Control experiments are not properly described in the main paper, only a few information are provided in the Supplementary Information. It is not clear how a specific absorption can represent a control measurement in this setup. A direct immobilization of biotin-streptavidin would be more significant, as authors can demonstrate the Debye screening would happen in that case.

(Reply)

We have already cited Ref. 24, in which the Debye screening effect on the biotin–streptavidin interactions was reported. The reference showed that the electrical signal of nanowire-FET devices was suppressed by the Debye screening in 1X PBS, compared with that in the diluted buffer 0.01X PBS. Considering this reference, we have added the following sentences in the revised manuscript.

“Indeed, the Debye screening effect on the biotin–streptavidin interactions at the sensor substrate was investigated previously [24], which showed that the electrical signal of nanowire-FET devices was suppressed by the Debye screening in 1X PBS, compared with that in the diluted buffer 0.01X PBS.” (page 7, line 258)

Authors state that their modification of the MCC approach, i.e. the magnetic anchoring of beads before biotin-streptavidin interaction, is more effective than two-stage magnetic bead approach. Why don’t reporting a demonstration of this?

(Reply)

As mentioned in the original manuscript, we have concerns about the previous MCC method, as follows.

“Background noise generated by the injection of magnetic beads, which corresponds to the effect of the direct attachment of magnetic beads at the gate, distorts the actual signal used for biomolecular recognition. This phenomenon occurs twice, before and after the reaction with the target biomolecules (Figure 1b).”

Therefore, we have added Figure S1 in Supplementary Materials as the representative results obtained by the previous method (Figure 1b). Also, we have added the following sentences regarding Figure S1.

Figure S1shows the change in interfacial potential (ΔVout) obtained by the previously reported MCC method (Figure 1b). Upon adding the biotin-coated magnetic beads with streptavidin, ΔVoutdrastically increased owing to the direct attachment of magnetic beads to the gate insulator and subsequently decreased gradually when the measurement solution flowed. That is, the effect of the direct attachment of magnetic beads to the gate insulator on the electrical responses was very large and then it took a long time to determine the baseline of interfacial potential; therefore, the actual signal used for biomolecular recognition may be distorted in the MCC method shown in Figure 1b.” (Supplementary Materials)

Authors would pay attention to the fact that their LOD (limit of detection, and NOT limitation on detection) is HIGHER than those reported in liturature, and not lower. They are considering this definition in an opposite way than the one normally accepted in literature. Moreover, 4 references are sufficient in the very wide range of publication about immunoFETs, and also the “Kaiser method” reported in the paper would be described, at least in SI as the related reference is in german!

(Reply)

In fact, the calculated LOD was inferior to those obtained in previous studies (Supplementary Materials S5), as described in the manuscript. However, the detection of biomacromolecules interacting with receptor molecules tethered at the gate insulator of an ISFET is not possible owing to the Debye length limit [Refs. 7–9]. That is, the signals detected by the MCC method provide the possibility of directly detecting biomacromolecules beyond the Debye length limit using an ISFET sensor. As we mentioned in the manuscript, we are going to improve the LOD by controlling the density of probe molecules (etc., biotin) at the magnetic beads and so forth in the future.

Regarding the Kaiser method, we have added the following sentences in the revised Supplementary Materials in accordance with the reviewer’s comment.

“S4. Calculation of LOD

Considering the Kaiser method, the lowest concentration of streptavidin that can be detected by the MCC method is calculated from  showing a significant difference from the average ΔVat the concentration of streptavidin in the blank (C= 0). Cindicates the concentration of streptavidin. In Figure 5, an approximately straight line is drawn in the concentration range from 1.8 μM to 180 μM, within which streptavidin can be detected, and extrapolated to the ΔVaxis. This relationship is expressed as

ΔV2= 3.7C+ 0.93 ,     (1)

where the y-intercept (0.93) means the average ΔVin the blank. Here, the corrected sample standard deviation (σ) is

where  shows the average  of a detected signal and nindicates the number of detected output signals. Moreover, the reliable range of win the blank is indicated as 

w = 3σ + 0.93.    (3)

That is, the LOD (CLOD) is calculated using equations (1)–(3)on the basis of the output signal  that shows +3σ from the average ΔVin the blank (0.93). 

Considering the above, the LOD for streptavidin for the MCC method was calculated to be about 2.3 μM.” (Supplementary Materials)

The explanation of Figure 4b is not clear. If streptavidin has a pI of 5.6, it should protonate at pHpI. Therefore, if pH is swept from acid to basic values, the positive charge would increase, and so the n-type ISFET would be switched off. Also considering an indirect detection through pH changes in the bead/gate insulator nanogap, this would be monotonic. So, it is not clear why ΔVis first decreasing (from 4.01 to 7.41) and then increasing (from 7.41 to 10.01). The explanation provided in Section 3.3 is confused and, according to the Reviewer, is more an hypothesis (badly discussed) than a proof.

(Reply)

As the reviewer pointed out, ΔVis expected to monotonically change with the pH of the measurement solution. We have considered the reason for this observation and the accurate control of the pH of buffer solutions. In fact, we need to control the ionic strength of buffer solutions even when the pH dependence of electrical responses is investigated. That is, we should use measurement solutions with almost the same ionic strength but different pHs. On the other hand, we have already shown the effect of ionic strength on the electrical responses using PBSs with near neutral pHs (Figure 4a). In fact, the ionic strength of 0.1X PBS was almost the same as that of the standard buffer solutions of pHs 4.01 and 10.01. Therefore, we consider that it is enough to compare and show the difference between the electrical responses at pHs 4.01 and 10.01. Considering the above, we have modified Figure 4b and added the following sentences in the revised manuscript.

“[pHs 4.01 (50 mM C6H4(COOK)(COOH)) and 10.01 (25 mM NaHCO3and 25 mM Na2CO3)]” (page 3, line 95)

“close to a neutral pH (pH 6.0 to 7.4)(page 4, line 168)

“Here, measurement solutions with almost the same ionic strength but different pHs should be used.” (page 5, line 194)

Too much emphasis is put in the fact that the ISFET has been integrated with a microfluidic. It is not the first time, and the proposed microfluidic is something very standard (I can say also very rudimentary).

(Reply)

As the reviewer commented, microfluidic systems are often used for biosensors. However, not only the ISFET sensors but also the MCC system was integrated in a microfluidic system in this study. This is a specific construction of our own device, which is not found in other works. We would be grateful if you could appreciate the concept of our device.

- Open Review

(x) I would not like to sign my review report 

( ) I would like to sign my review report 

- English language and style

(x) Extensive editing of English language and style required 

( ) Moderate English changes required 

( ) English language and style are fine/minor spell check required 

( ) I don't feel qualified to judge about the English language and style 

Yes

Can be improved

Must be improved

Not applicable

- Does the introduction provide sufficient background and include all relevant references?

Can be improved

- Is the research design appropriate?

Must be improved

- Are the methods adequately described?

Must be improved

- Are the results clearly presented?

Must be improved

- Are the conclusions supported by the results?

Must be improved

Round 2

Reviewer 3 Report

Dear Editor,

I went throug the revised version of the manuscript “Molecular-Charge-Contact-Based Ion-sensitive Field-Effect Transistor Sensor in Microfluidic System for Protein Sensing”. Unfortunately, I didn’t found the manuscript improved enough to receive my endorsement for publication. On the contrary, my concerns in authors’ approach and presentation get worse.

My biggest concern is related to the fact that authors are employing, as a reference for their experiments, a previous study cited in the text (Ref. 24). Although this paper focuses on Debye screening lenght effects, this is NOT an ISFET structure. Screening effects are related not only to ionic strenght of the solutions, but also to electrict field in the measurement environment, which is very different between an ISFET and a NW-FET, as the latter does not have a reference electrode imposing the potential in the measurement solution. In this sense, using this result as reference is misleading. It is my opinion an evaluation of ISFET response when biotin are chemically anchored on the gate insulator is mandatory to evaluate the actual advantages of the approach proposed by authors.

Another problem is related to Fig. 4b and authors’ response to my concerns (related to the fact that device response to different pH value is not monotonic). It is not clear why authors have not the possibility to control pH value and ionic strength at the same time. In any case, despite what reported by authors, two pH points are not enough to represent the validity of the device response to pH variations, and indeed represents a sub-standard for scientific publication. If authors want to characterize device response to pH, they have to employ, in such a wide pH range, at least 5-6 different points.

Finally, I would like to stress about that LoD definition is wrong. As LoD is generally describe as the lower amount of analyte that can be recognized with respect to the noise, authors are demonstrating an HIGHER LoD than literature, not an inferior LoD. This must be changed, as it is not representative of the commonly used LoD definition. Moreover, I highlight once more than 5 references for comparison are not sufficient.

The introduction in SI of control experiments for MCC approaching after biotin-streptavidin interaction and LoD calculations is fine. Nonetheless, I didn’t find the level of English presentation improved.

Summarizing, I still don’t evaluate the overall innovation and quality of presentation sufficient for publication.

Author Response

Dear Editor:

Thank you very much for your kind reviews. We have revised our manuscript in accordance with the reviewers’ comments as follows. We would be grateful if the manuscript could be re-valuated for publication in Sensors.

[Reviewer 3]

Comments and Suggestions for Authors:

I went throug the revised version of the manuscript “Molecular-Charge-Contact-Based Ion-sensitive Field-Effect Transistor Sensor in Microfluidic System for Protein Sensing”. Unfortunately, I didn’t found the manuscript improved enough to receive my endorsement for publication. On the contrary, my concerns in authors’ approach and presentation get worse.

My biggest concern is related to the fact that authors are employing, as a reference for their experiments, a previous study cited in the text (Ref. 24). Although this paper focuses on Debye screening length effects, this is NOT an ISFET structure. Screening effects are related not only to ionic strength of the solutions, but also to electric field in the measurement environment, which is very different between an ISFET and a NW-FET, as the latter does not have a reference electrode imposing the potential in the measurement solution. In this sense, using this result as reference is misleading. It is my opinion an evaluation of ISFET response when biotin are chemically anchored on the gate insulator is mandatory to evaluate the actual advantages of the approach proposed by authors.

(Reply)

Thank you very much for the careful review. However, we beg to disagree with the reviewer’s comments. The Debye screening effect is generally found at an electrolyte solution/sensing electrode interface of potentiometric sensors. That is, even if biomolecular recognition events are detected at the sensing electrode of nanowire–FET devices, the Debye screening effect should be considered as the detection principle, which is similar to that of ISFET sensors in a broad sense. In particular, the study in Ref. 24 showed the Debye screening effect on the streptavidin–biotin interaction by varying the concentration of PBS from 0.01 to 1X, as similarly observed in our present study. Therefore, we believe that our previous reply regarding the Debye screening effect is not misleading.

In addition, the most important point is that we consider how the buffer concentration of the measurement solution should be controlled to avoid the Debye screening effect. Regarding Ref. 24, the same authors used 1 mM sodium bicarbonate buffer solution (pH 8.4), whose ionic strength was near that of the 0.01X PBS solution, for streptavidin–biotin sensing to avoid the Debye screening effect, as reported in a previous paper [Ref. 23]. However, the drift of electrical signals in the 0.01X PBS solution may not have been negligible owing to the deteriorating buffer effect, considering our experiments (and experiences). That is, the 0.1X PBS solution was employed for the real-time measurement (Figure 3) and LOD analysis (Figure 5). Therefore, we have added the following sentences to support our previous reply.

“Therefore, the 1 mM sodium bicarbonate buffer solution (pH 8.4), whose ionic strength was near that of the 0.01X PBS solution, was utilized for streptavidin–biotin sensing to avoid the Debye screening effect reported in a previous paper [23]. However, the drift of electrical signals in the 0.01X PBS solution may not have been negligible owing to the deteriorating buffer effect. That is, the 0.1X PBS solution was employed for the real-time measurement (Figure 3) and LOD analysis (Figure 5) in this study. Moreover, to avoid the Debye screening effect,” (page 7, line 259)

Another problem is related to Fig. 4b and authors’ response to my concerns (related to the fact that device response to different pH value is not monotonic). It is not clear why authors have not the possibility to control pH value and ionic strength at the same time. In any case, despite what reported by authors, two pH points are not enough to represent the validity of the device response to pH variations, and indeed represents a sub-standard for scientific publication. If authors want to characterize device response to pH, they have to employ, in such a wide pH range, at least 5-6 different points.

(Reply)

As we mentioned in our previous reply, we consider that it is sufficient to compare and show the difference between the electrical responses at pHs 4.01 and 10.01 (Figure 4b) in this paper. In fact, we need to control the ionic strength of buffer solutions even when the pH dependence of electrical responses is investigated. That is, we should use measurement solutions with almost the same ionic strength but different pHs. On the other hand, we have shown the effect of ionic strength on electrical responses using PBS solutions with near-neutral pHs (Figure 4a). The ionic strength of 0.1X PBS with pH 6.6 was almost the same as that of the standard buffer solutions with pHs 4.01 and 10.01. Also, we have added the components of the standard buffer solutions with pHs 4.01 and 10.01 in the revised manuscript in accordance with the reviewer’s previous comment. Considering the above, we highlight the following sentences in the revised manuscript again. We would be grateful if the reviewer could understand our viewpoint.

“pH 6.6 for the original 0.1X PBS solution” (page 4, line 157)

“PBS solution from 1X to 0.01X close to a neutral pH (pH 6.0 to 7.4)” (page 4, line 167)

Finally, I would like to stress about that LoD definition is wrong. As LoD is generally describe as the lower amount of analyte that can be recognized with respect to the noise, authors are demonstrating an HIGHER LoD than literature, not an inferior LoD. This must be changed, as it is not representative of the commonly used LoD definition. Moreover, I highlight once more than 5 references for comparison are not sufficient.

(Reply)

In accordance with the reviewer’s comment, we have reconsidered the references on the limit of detection (LOD) and added some references in Table S1 and the revised manuscript as follows.

“26. Gebauer, C.R.; Rechnitz, G.A. Ion Selective Electrode Estimation of Avidin and Biotin Using a Lysozyme Label. Anal. Biochem. 1980, 103, 280–284. doi: 10.1016/0003-2697(80)90610-7

“27. Mock, D.M.; Langford, G.; Dubois, D.; Criscimagna, N.; Horowitz, P. A Fluorometric Assay for the Biotin-Avidin Interaction Based on Displacement of the Fluorescent Probe 2-Anilinonaphthalene-6-sulfonic Acid. Anal. Biochem. 1985, 151, 178–181. doi: 10.1016/0003-2697(85)90068-5”

“28. Schray, K.J.; Artz, P.G.; Hevey, R.C. Determination of Avidin and Biotin by Fluorescence Polarization. Anal. Cham. 1988, 60, 853-855. doi: 10.1021/ac00160a006”

“29. Wu, Y.; Ma, H.; Gu, D.; He, J. A Quartz Crystal Microbalance as a Tool for Biomolecular Interaction Studies. RSC Adv. 2015, 5, 64520–64525. doi: 10.1039/c5ra05549k”

On the other hand, we need to fairly show the present situation on the LODs with respect to the streptavidin–biotin interaction that were reported in previous papers, regardless of whether it is good or bad. In particular, considering our result for the LOD, we pointed out the possibility or merit of our method in the original manuscript as follows. “However, the detection of biomacromolecules interacting with receptor molecules tethered at the gate insulator of an ISFET is not possible owing to the Debye length limit [7–9]. That is, the signals detected with the MCC method provide the possibility of directly detecting biomacromolecules beyond the Debye length limit using an ISFET sensor, although the LOD should be improved in the future. (page 6, line 211)” In addition, we have found a few references where the LODs were relatively equal to that in our paper; therefore, we have revised the sentence on the calculated LOD, as follows.

“The calculated LOD is equal or inferior to those obtained in previous studies, as shown in Table S1 (Supplementary Materials S5) [26–30].” (page 6, line 210)

The introduction in SI of control experiments for MCC approaching after biotin-streptavidin interaction and LoD calculations is fine. Nonetheless, I didn’t find the level of English presentation improved.

(Reply)

In accordance with the reviewer’s comment, we have had the revised manuscript checked and proofread by a native English speaker. We would be grateful if the manuscript could be re-evaluated for publication in Sensors.
